# Nebulized Glutathione as a Key Antioxidant for the Treatment of Oxidative Stress in Neurodegenerative Conditions

**DOI:** 10.3390/nu16152476

**Published:** 2024-07-31

**Authors:** João Vitor Lana, Alexandre Rios, Renata Takeyama, Napoliane Santos, Luyddy Pires, Gabriel Silva Santos, Izair Jefthé Rodrigues, Madhan Jeyaraman, Joseph Purita, Jose Fábio Lana

**Affiliations:** 1Medical School, Max Planck University Center (UniMAX), Indaiatuba 13343-060, SP, Brazil; jvblana@gmail.com (J.V.L.); josefabiolana@gmail.com (J.F.L.); 2Department of Orthopedics, Brazilian Institute of Regenerative Medicine (BIRM), Indaiatuba 13334-170, SP, Brazil; alemrrios@gmail.com (A.R.); renata.takeyama@gmail.com (R.T.); dranapolianesantos@gmail.com (N.S.); luyddypires@gmail.com (L.P.); neurovirtual@gmail.com (I.J.R.); 3Regenerative Medicine, Orthoregen International Course, Indaiatuba 13334-170, SP, Brazil; madhanjeyaraman@gmail.com (M.J.); jpurita@aol.com (J.P.); 4Department of Orthopedics, ACS Medical College and Hospital, Dr MGR Educational and Research Institute, Chennai 600095, India; 5Medical School, Jaguariúna University Center (UniFAJ), Jaguariúna 13918-110, SP, Brazil; 6Clinical Research, Anna Vitória Lana Institute (IAVL), Indaiatuba 13334-170, SP, Brazil

**Keywords:** glutathione, nebulization, oxidative stress, neurological disorders, aging, regenerative medicine

## Abstract

Glutathione (GSH), a tripeptide synthesized intracellularly, serves as a pivotal antioxidant, neutralizing reactive oxygen species (ROS) and reactive nitrogen species (RNS) while maintaining redox homeostasis and detoxifying xenobiotics. Its potent antioxidant properties, particularly attributed to the sulfhydryl group (-SH) in cysteine, are crucial for cellular health across various organelles. The glutathione-glutathione disulfide (GSH-GSSG) cycle is facilitated by enzymes like glutathione peroxidase (GPx) and glutathione reductase (GR), thus aiding in detoxification processes and mitigating oxidative damage and inflammation. Mitochondria, being primary sources of reactive oxygen species, benefit significantly from GSH, which regulates metal homeostasis and supports autophagy, apoptosis, and ferroptosis, playing a fundamental role in neuroprotection. The vulnerability of the brain to oxidative stress underscores the importance of GSH in neurological disorders and regenerative medicine. Nebulization of glutathione presents a novel and promising approach to delivering this antioxidant directly to the central nervous system (CNS), potentially enhancing its bioavailability and therapeutic efficacy. This method may offer significant advantages in mitigating neurodegeneration by enhancing nuclear factor erythroid 2-related factor 2 (NRF2) pathway signaling and mitochondrial function, thereby providing direct neuroprotection. By addressing oxidative stress and its detrimental effects on neuronal health, nebulized GSH could play a crucial role in managing and potentially ameliorating conditions such as Parkinson’s Disease (PD) and Alzheimer’s Disease (AD). Further clinical research is warranted to elucidate the therapeutic potential of nebulized GSH in preserving mitochondrial health, enhancing CNS function, and combating neurodegenerative conditions, aiming to improve outcomes for individuals affected by brain diseases characterized by oxidative stress and neuroinflammation.

## 1. Introduction

In 1888, Joseph Charles François de Rey-Pailhade discovered the glutathione molecule but it was only in 1929, through the efforts of Gowland Hopkins, Hunter, and Eagles, that it was possible to establish the composition of this molecule [1,2].

Glutathione (GSH) is a tripeptide molecule comprising cysteine, glycine, and glutamate. GSH is an important antioxidant found extensively throughout the body and synthesized in the cytosol. GSH neutralizes reactive oxygen species (ROS), reactive nitrogen species (RNS), and electrophiles directly via enzymatic reactions. It also plays a role in maintaining redox homeostasis, regulating cellular events, and detoxifying xenobiotics [3].

GSH possesses potent antioxidant properties (Table 1) due to the presence of the sulfhydryl or thiol (-SH) functional group from the cysteine (Cys) amino acid [3]. GSH is the main low-molecular-weight thiol found in living cells and is also considered one of the major non-protein defenders against oxidative stress (OS) [4]. After its synthesis in the cytosol, it is transported and distributed to different intracellular organelles, such as the mitochondria, endoplasmic reticulum, and nucleus [2,3]. GSH works in coordination with other redox compounds, such as nicotinamide adenine dinucleotide phosphate (NADPH), helping regulate and maintain the cellular redox status [2,3]. The mechanism of action of GSH involves two key enzymes: glutathione peroxidase (GPx) and glutathione reductase (GR). GPx is responsible for converting GSH to its oxidized form, producing oxidized glutathione disulfide (GSSG), while GR reduces GSSG back to GSH, establishing an enzymatic cycle regenerating GSH from GSSG [2,3]. GSH is also an essential cofactor for enzymes that recycle other endogenous antioxidants like vitamins C and E [5].

The GSH-GSSG cycle in its primary function helps cells to remove hydrogen peroxide (H_2_O_2_) and lipid peroxides, converting them to water and lipid hydroxides. The detoxification process of H_2_O_2_ to H_2_O imparts a protective role to the cell, promoted by the hijack of 20% of O_2_ in the body, reducing ROS and oxidative stress, thus decreasing cellular and molecular oxidative damage [3,5,6]. Through these mechanisms, the glutathione cycle helps mitigate oxidative damage in various tissues, reducing the free radicals produced by cellular metabolism, diminishing the generation of ROS and RNS, and attenuating inflammation [2,3].

Considering ROS and its contributions to tissue damage, the mitochondrion plays a significant role in ROS formation as it is a major producer of these particles, especially superoxide radicals (-O_2_) [2,3,7]. The GSH-GSSG cycle, however, seems to play a crucial role in mitigating mitochondrial oxidative damage. This cycle is essential for autophagy, apoptosis, and ferroptosis, thus being involved in the regulation of metal homeostasis, preventing the formation of toxic metal complexes by reducing redox-active metals like iron and copper [2,3,5]. GSH concentrations are required for many physiological processes, such as the activation of T-cells, cytokine production, and modulation of the immune response [4].

Many studies have outlined the role of oxidative stress in affecting the nervous system and its function through distinct mechanisms. The central nervous system is particularly vulnerable to oxidative stress and damage due to its high consumption of oxygen, the rich content of polyunsaturated fatty acids, and the relatively low levels of antioxidants [8,9].

This review explores the role of glutathione in human health and its possible contributions to the protection of neuronal components, restoration of neurological function, and regulation of oxidative stress in the central nervous system.

## 2. Methods

A comprehensive literature search was conducted across multiple electronic databases, including PubMed, Google Scholar, Cochrane Library, SCIELO, and LILACS, to identify relevant scientific data on the role of glutathione in neurological conditions related to oxidative damage and the potential of nebulized GSH as a therapeutic approach. Only studies related to glutathione and neurological oxidative stress written in English with access to the full text were considered for this review.

The search strategy employed a combination of keywords and Medical Subject Headings (MeSH) terms related to glutathione, oxidative stress, neurodegenerative disorders (including Parkinson’s Disease and Alzheimer’s Disease), antioxidant mechanisms, and nebulization. Keywords included “glutathione”, “GSH”, “oxidative stress”, “neurodegenerative diseases”, “Parkinson’s Disease”, “Alzheimer’s Disease”, “nebulization”, “antioxidant therapy”, and “NRF2 pathway”.

## 3. Glutathione Biochemistry

GSH biosynthesis occurs in the cytosol, requiring the participation of the endoplasmic reticulum (ER), mitochondria, nuclear matrix, and peroxisomes [4,10]. The key determinants of GSH synthesis are cysteine and the activity of rate-limiting enzyme, glutamate cysteine ligase (GCL). The GCL is composed of a catalytic (GCLC) and a modifier (GCLM) subunit. GCL catalyzes the formation of y-glutamyl-cysteine from glutamate and cysteine, initiating the first step of GSH synthesis. The next step is performed by GSH synthase (GS), which catalyzes the formation of GSH from y-glutamyl-cysteine (y-GCS) and glycine [5,7,11,12]. ATP hydrolysis is necessary for this system to work properly, therefore implying a significant involvement of the mitochondria [2,13].

Cysteine (Cys) supply is derived from diet, particularly from protein catabolism and trans-sulfuration of methionine. This is the pathway that the sulfur-containing amino acid enters the biosynthesis of GSH via homocysteine (HCys) formation. Extracellularly, the Cys molecule is not stable and it rapidly auto-oxidizes to cystine (CysSS). CysSS is recruited by several cells being converted again to Cys intracellularly. This is referred to as the Cysteine-Cystine (Cys-CysSS) cycle [4,10,11].

In order to form GSH, two main molecules are needed: glutamate (Glu) and cysteine (Cys). They are linked through a y-carbonyl group of Glu and this link confers GSH with a high molecular stability. Since the GSH molecule has strong stability, it is able to perpetuate the cycle by relying on some enzymes that are capable of regulating its degradation. The y-glutamyl-transpeptidase (GGT) catalyzes the hydrolysis of this y-carbonyl group of Glu, generating Cys-Gly [4,5,10].

In the GSH system, a few key genes are involved in biosynthesis, such as y-GCS subunits and CysSS transponders, regulated at transcriptional levels by the same genes that also control GSH-dependent detoxification genes, like glutathione peroxidase (GPx) and glutathione S-transferase (GST). It appears that the thioredoxin (Trx) system is also associated with GSH at a transcriptional level. Another redox-sensitive transcription factor seems to have an affinity for certain inducible antioxidant genes. These redox-sensitive transcription factors include the “Cap’n’Collar” family. Nuclear factor erythroid 2 (NFE2)-related factor (NRF2) belongs to this transcription factor family, being by far the most assorted investigated transcriptional isoform factor. The NRF2 role is well established in the activation of the antioxidant system and GSH expression preservation [3,4,10,11,14].

## 4. NRF2 Pathway

GSH enzymes are regulated at multiple levels and often in a coordinated manner. The key transcription factors that regulate the expression of GCL subunits and GSH synthase include nuclear-related factor 2, via the antioxidant response element (ARE), activating protein-1 (AP-1), and nuclear factor kappa B (NFkB) [2,13,15]. NRF2 is activated by a cellular electrophile or by a protein kinase-mediated phosphorylation, migrating into the nucleus and dimerizing itself with Maf proteins binding to specific DNA sequences [4,15].

NRF2 acts by binding to the antioxidant response element in the promoter region of these genes activating their transcription. In organisms kept under non-stressful physiological conditions, NRF2 is kept in the cytosol by Keap1, a component of an E3 ubiquitin ligase complex that targets NRF2 for proteasomal degradation [15,16,17,18]. To clarify, NRF2 is a member of the “Cap’n’Collar” (CNC) transcription factor family, which consists of 605 amino acids and is divided into seven highly conserved functional domains known as Neh1-Neh7. All these domains help maintain the stability of ubiquitination, ubiquitin conjugation, and nuclear translocation signaling of NRF2, being responsible for DNA-binding and transactivation-mediated interaction of NRF2 with co-factors [4,10,18,19].

Keap1, consisting of 624 amino acids, is a cysteine-rich protein containing 27 cysteine residues in humans. Of these 27 cysteine residues, C151, C273, and C288 are highly susceptible to reactions of covalent modifications promoted by ROS, RNS, and other electrophiles. Some processes affecting critical residues include S-sulfenylation, S-nitrosylation, and S-sulfhydration that induce conformational changes in Keap1. This change in Keap1 promotes the dissociation of NRF2 and its stabilization [9,20,21,22].

Once NRF2 is released from Keap1, it escapes proteasomal degradation and is translocated to the nucleus, inducing the activation of genes involved in antioxidant defense [4,16,23,24,25]. NRF2 activation leads to increased expression of GSH synthesis enzymes such as GCL and GSH recycling enzymes, such as GR [2,11,12].

The literature has identified a binary interaction of NRF2 in the oxidant-activated adaptative stress response mediated and controlled by the NRF2/GSH axis. This response is associated with Cysteine residues from Keap1 and the role of GSH in redox control of thiols [4,5,22,24].

This redox mechanism consists of the ability of cysteine residues to undergo reversible changes in their redox state, modifying their protein conformation and catalytic and regulatory functions. Sulfhydryl groups derived from Cys residues can be present as thiols, disulfides, or mixed disulfides. These Cys residues can be conjugated with GSH forming protein S-Glutathionylation (P-SSG) [5,12,22,24,25]. These protein-mixed disulfides often exist under oxidative and nitrosative stress conditions. Indeed, P-SSG is involved in protein kinases, checkpoints of cell-cycle regulation, and cell death pathways, which are all involved in the increase in ROS and RNS in a controlled manner [4,5,24,25]. P-SSG appears to exist in proteins that participate in energy metabolism regulation [4,14,22,23].

Glutathione S-transferases, particularly the P isoform (GSTP) and the glutaredoxins (Grx), are credited as the main proteins in the enzymatic S-Glutathionylation of cellular proteins. This P isoform regulates the production of S-Glutathionylation proteins in cells increasing or decreasing the cellular exposure to redox stressors and glutathionylating agents. The substrate that forms the GSTP-dependent S-Glutathionylation function was identified as peroxiredoxin 6 (Prdx6) [4,5,14,22,23]. Prdx6 is a 1-Cys peroxidase that has a reductive activity related to membrane hydroperoxide constituents. In order for this reductive ability to work well, GSTP genes are required for redox restoration. This redox activity relies on GSH and catalytic Cys of Prdx6, a process in which GSTP directly interacts with GSTP and Prdx6, transferring the GSH molecule to its 1-Cys. The final step is a thiolase reaction that completes the redox cycle restoration of the catalytic Cys [4,10,22,23,24,25]. The expression of the GSTP gene indicates an essential role in the control of the NRF2 activity by downregulation when the GSTP gene is induced, helping complete the adaptive stress response [4,5,10,19,22,23,25].

Studies about the cycle of GSH and the role of NRF2 have shown that GSH can be recycled from its oxidized state by the enzyme glutathione reductase [2,7,15]. These studies have found that GSH recycling is critical for cell survival under conditions of oxidative stress. NRF2 regulates GR expression and GSH recycling regardless of GSH biosynthesis. The literature also shows that knockdown of GR expression or inhibition of this enzyme activity can make cells more sensitive to oxidative stress-induced cell death. The studies suggest that GSH recycling is an essential component of the cellular antioxidant system and that NRF2-mediated GR expression and regulation is one of the most important mechanisms for maintaining cellular redox homeostasis and cell survival under conditions of oxidative stress [4,5,11,14,15,22]. Table 2 provides a more concise summary of the interaction between GSH and the NRF2 pathway.

### 4.1. Glutathione in NRF2 Signaling and Neurological Disorders

When NRF2 presents a dysfunction, the antioxidant system fails, causing decreased activity in the GSH cycle, which increases oxidative stress and damage. NRF2 dysfunction has been implicated in various neurodegenerative diseases, especially Alzheimer’s Disease (AD) and Parkinson’s Disease (PD), where oxidative stress plays a significant role in disease pathogenesis. Evidence indicates that the NRF2 pathway is impaired in neurodegenerative conditions, rendering neurons vulnerable to oxidative damage [2,3,7,26,27,28,29,30,31,32,33,34,35]. Table 3 summarizes the role of glutathione in the pathological features of PD and AD.

The brain has the majority of its GSH (97%) in the reduced form and its levels are crucial for maintaining neuronal health and preventing cell and mitochondrial degeneration. In the central nervous system, the highest concentration of GSH is found in the glial cells of the cortex. The highest concentration regions of GSH in the brain, after the cortex, are observed in the hypothalamus, cerebellum, striatum, and substantia nigra (SN). GSH is also present in both the extracellular fluid and the cerebrospinal fluid (CSF) [6,7,36]. The GSH absorption is highest in the retina, compared to other regions. Interestingly, however, the detoxification of compounds by GSH coupling occurs primarily in the kidneys and the liver but not in the brain itself [6,7,36,37].

The metabolic interaction between neurons and astrocytes is critical for glutathione synthesis. In vitro, studies with co-cultured neurons and astrocytes showed that GSH levels in neurons increase in the presence of astrocytes. This happens because the astrocyte has an associated membrane with gamma-glutamyl transpeptidase (yGT) that converts extracellular GSH to the dipeptide CysGly and subsequently to Cys and Gly. This transfers the cysteine (Cys) precursor to neurons, upregulating the neuronal GSH synthesis [2,3,11,23,38,39,40]. GSH synthesis, promoted by the interaction between neurons and astrocytes, is dependent on the glutamate-glutamine-cycle, providing Glu peptide to combine with the CysGly dipeptide in the extracellular space. Glu is carried to astrocytes via excitatory amino acid carrier 1 (EAAC1), allowing the GSH synthesis. The neuron-astrocyte interactions transport intracellular GSH into the extracellular space via multiple drug resistance protein 1 (MRP1), hydrolyzing it into CysGly by the y-GT enzyme [3,4,6,7,23,37,38,39]. Many cells and tissues release GSH into the circulatory system, which promotes its transfer between cells. The plasma levels of total glutathione (GSH + GSSG) are equivalent to approximately 85% GSH and 15% GSSG. Several organs absorb GSH from the plasma by direct uptake of GSH or by the breakdown of GSH by y-GT, even though there is a specific transport system of Cys amino acids that compete with other plasma amino acids for L-system transport across the blood–brain barrier [2,4,6,7,23,37,38,39,41].

The central nervous system has a very high demand for O_2_ in order to operate adequately. The brain represents about 2% of the body weight and, even with this disproportional mass balance, it can consume up to 20% of the total O_2_ distributed throughout the whole body. The blood–brain barrier (BBB) is formed by the blood vascular endothelial cells lining the cerebral microvessels and is essential for the control of the brain microenvironment by separating the blood from the brain. The BBB has different routes to receive various molecules [3,26,37]. Many of these routes are supported by tight junctions, called paracellular transport. This route is responsible for the transportation of simple water-soluble molecules. Passive diffusion is another possibility for molecular transport, carrying lipid-soluble substances like steroids and alcohol. Amino acids and glucose are transported via solute carrier-mediated transport based on a concentration gradient route using ATP as an energy source [3,6,37].

Macromolecules can cross the BBB via receptor-mediated transcytosis and adsorptive-mediated transcytosis. GSH acts like a tripeptide across the BBB through sodium-dependent transport of the GSH transponder that belongs to the solute carrier family. The sodium-dependent GSH transponders are localized in the lumen of endothelial cells and in astrocyte membranes. This transponder has also been shown to be bidirectional, meaning that it does the uptake and reuptake of GSH in BBB [3,6,7,36].

With the aging process, GSH plays a vital role in neural defense against the oxidative damage caused by oxidants like ROS and RNS. The non-neuronal elements of CNS and peripheral nervous system (PNS), namely the glia, ependymal, and endothelia, show high levels of GSH that are not found in neuronal cell bodies or granule cells. The GST enzyme is well distributed in the CNS in many subunits. This enzyme catalyzes the conjugation of GSH with xenobiotic compounds that yield nontoxic products, eliciting detoxification [7,26,37].

The astrocytes and oligodendrocytes express two subunits of GST: u-GST and n-GST. The pial, ependymal, and vascular elements also express many quantities of these GST subunits. Another subunit produced in the CNS is the Alpha-GST present in neurons and non-neuronal compounds. In this sense, the GSH system has a pivotal role in the regulation of redox homeostasis since the stages of embryogenesis. The developing nervous system is exposed to various xenobiotics through the placental circulation during fetal development, suffering a dramatic change in oxygenation at birth when newborns are exposed to the hyperoxic extrauterine scenario [6,7,26,37].

Therefore, it is imperative to maintain homeostasis of the GSH levels principally during the aging process. The aging process is mainly characterized by a gradual decline of the body’s biological functions, which can lead to increased production of ROS [9,37]. If ROS generation exceeds the capacities of the antioxidant system to neutralize them, a damaging state of oxidative stress may be established. The antioxidant system plays a key role in maintaining the balance between oxidation and reduction and GSH is the most abundant antioxidant present in human cells. For this reason, variations in GSH levels, especially when decreased, can be considered the cause of oxidative stress and cell damage [26,37].

### 4.2. Electron Transport Chain

In order to understand the variations in GSH levels in the body, it is also necessary to comprehend the GSH electron transport. To perform well, human cells evolved to develop robust homeostatic mechanisms to keep safe against oxidation or alkylation by electrophilic species [10,11]. As was discussed before, mitochondria have an intrinsic relationship with the GSH cycle. In mammalian cells, mitochondria have GSH storage, which accounts for approximately 10–15% of total GSH in cells. This organelle generates most of the cellular energy by means of oxidative phosphorylation (OXPHOS), which is fundamental for the maintenance of many cellular functions. OXPHOS provides a mechanism that couples electron transport with the synthesis of ATP from ADP. Above this mechanism, mitochondria are also involved in key cellular functions like calcium homeostasis, biosynthesis of the heme group, steroid hormones, metabolism of nutrients, and removal of ammonia [11,12,42]. Mitochondria are also integrated in metabolic and signaling pathways of apoptosis and autophagy [43].

Evidence indicates that mitochondria are involved in a central role of initiating the signals to respond to metabolic and genetic stress, impacting nuclear gene expression and leading to changes in cellular function [44,45].

OXPHOS is organized in a series of steps that include several redox centers distributed in five protein complexes in the inner mitochondrial membrane (IMM) [44,45,46].

Complex I: Also known as the ubiquinone oxidoreductase pathway, it oxidizes NADH from glycolysis and the citric acid cycle, transferring electrons to coenzyme Q and pumping 4 hydrogen ions across the membrane. (Coenzyme Q: Acts as an electron carrier transferring electrons to complex III after undergoing reduction to CoQH2)

Complex II: The succinate dehydrogenase pathway accepts electrons from succinate in the citric acid cycle but does not pump protons across the membrane;Complex III: Composed of cytochrome b, Rieske subunits, and cytochrome c proteins, it transfers electrons from coenzyme Q to cytochrome c in a two-step process called the Q cycle;Complex IV: Known as cytochrome c oxidase, it transfers electrons from cytochrome c to oxygen, producing water and contributing to the proton gradient;ATP synthase (Complex V): Utilizes the proton gradient generated by the ETC to produce ATP by rotating F0 and F1 subunits.

These complexes work together in a coordinated manner to generate ATP efficiently through OXPHOS. In order for complex I to obtain electrons from NADH and complex II from succinate, both complexes rely on a lipid-soluble carrier of ubiquinone (coenzyme Q) to form ubiquinol [12,44,45]. With ubiquinol, the electrons shift the redox gradient through complex III, turning the ubiquinol to coenzyme Q-cytochrome c oxidoreductase, then to complex IV, where oxidation of coenzyme Q occurs. The fall in electron potential energy through this ETC is used to pump protons out of the mitochondrial matrix to the intermembrane space (IMS). This generates a proton-motive force that is used by the fifth protein complex with ATP synthase, regenerating ATP from ADP [12,44,45,46,47,48].

The primary function of mitochondria is to produce ATP. However, during the ETC, a small portion of the electrons are transferred directly to O_2_, resulting in superoxide anion generation, which can elevate the levels of free radicals. Therefore, they also inevitably become a major producer of ROS [12,45].

Mitochondria are the primary intracellular site of oxygen consumption; it has been estimated that the steady-state concentration of superoxide in the mitochondrial matrix is 5 to 10 times higher than in the cytosol. Normally, ROS generated under physiological conditions are not harmful. Conversely, under stressful conditions like hypoxia, ischemia/reperfusion injury, chemical stress, and drug administration, mitochondrial ROS generation increases and plays a signaling role. This elevation gives rise to oxidative stress and initiates inflammation triggered by oxidation-mediated cell damage due to the accumulation of superoxides [12,45].

### 4.3. Glutathione Support of Mitochondrial Function

Electron transport proteins, such as cytochrome C oxidase and Complex I, are particularly vulnerable to oxidative damage because they are directly involved in the generation of free radicals during the ETC. GSH plays an essential role in mitigating OS and damage to electron transport proteins by acting as a potent antioxidant, providing indispensable support in mitochondrial function (Table 4). GSH can donate electrons to mitigate oxidation, avoiding damage to the cellular components, including electron transport protein complexes. When GSH donates an electron, it becomes GSSG; this molecule is catalyzed by GR, which reduces GSSG back to GSH using NADPH as an electron donor. This process also includes the recycling of coenzyme Q10, which is another key component of the ETC. The reduction of coenzyme Q10 is mediated by GSH, reducing the oxidized form of coenzyme Q10; this coenzyme is a lipid-soluble electron carrier that shuttles electrons between complexes I and II of the ETC [11,12,42,47,48].

GSH is the main intracellular thiol and is essential for the detoxification of electrophiles and oxidants. Homeostasis of this thiol is mediated by enzymatic or non-enzymatic oxidation of GSSG, succeeding NADPH-dependent reduction to GSH by GR. Since GSSG is not readily exported out of mitochondria, the activity of GR plays an indispensable role in controlling the level of intramitochondrial GSSG. The uncontrolled generation of GSSG during OS can contribute to mitochondrial dysfunction [10,11,42].

The first line of defense against the superoxides is given by the presence of a specific member of the family of metalloenzymes called superoxide dismutases (SODs) [5,11,12,42,49]. SOD2 is specifically located in the mitochondrial matrix, catalyzing the dismutation of superoxide anion into H_2_O_2_; however, distinct SOD enzymes (Cu, Zn-SOD, or SOD1) can eliminate the superoxides released in the IMS. The dismutation of superoxide by SOD2 is the predominant source of H_2_O_2_. Although H_2_O_2_ is not a free radical, per se, it is indeed a potent and harmful oxidant that can attack mitochondrial components like proteins, lipids, and DNA. Additionally, H_2_O_2_ can be a potential source of more reactive free radicals via the Fenton reaction. Normally, H_2_O_2_ is transported across membranes by aquaporins. The Fenton reaction is the conversion process of H_2_O_2_ in highly reactive hydroxyl radical (OH^−^) in the presence of transition metals, like Fe^2+^ and Cu^+^ [12,42,45,50,51,52].

The detoxification against H_2_O_2_ in the mitochondria occurs mainly through the GSH redox system, being executed by Gpx and GR as well as the presence of peroxiredoxins, using the reduced equivalents of NADPH. The supply of NADPH is essential to regenerate GSH, dictating the rate of H_2_O_2_ reduction by Gpx, keeping the mitochondria in a reduced state, and preserving ATP production [11,12,48,52].

Glutathione peroxidases have been identified in humans as eight isoforms, varying in cellular location and substrate specificity. Gpx1 is the major isoform localized in many cellular components, including mitochondrial matrix and IMS, being present in the liver and accounting for one-third of the total Gpx activity [2,53]. Gpx1 works in coordination with y-glutamylcysteine as a cofactor in mitochondrial H_2_O_2_ detoxification, mimicking the physiological properties of GSH. This peroxidase has been described as a selenium-containing homotetramer protein that has substrate specificity for H_2_O_2_ reducing enzyme. Gpx1 and Gpx4 display a preference for lipid hydroperoxides and hence play key roles in protecting phospholipids, cholesteryl esters, and cardiolipins. Gpx1 and Gpx4 participate in defense against apoptosis and maintenance of ETC and OXPHOS. In line with this vital role in mitochondrial defense, Gpx4 is more sensitive to oxidative stress triggers and has shown the ability to modulate the ferroptotic pathway, a specific form of cell death characterized by the production of iron-dependent ROS generation [2,5,12,31,44,45,50,51].

It is evident that, without the antioxidant system functioning under stable physiological conditions, the mitochondria will suffer dysfunctional changes. These dysfunctional changes can be impaired principally in the mitochondrial protein complexes. Once these protein complexes are not functioning well, the process involved in ATP generation can also be affected. Additionally, it is important to stabilize the mitochondrial microenvironment as these organelles play vital roles not only in energy production but also in other functions including regulation of cell death events. [2,5,11,12,37,41,42,43,44,46,50,52].

### 4.4. Apoptotic Pathways, Mitochondrial Dysfunction, and Oxidative Stress

Apoptosis or programmed cell death is characterized by a series of biochemical events that lead to cell fragmentation into compact membrane-enclosed structures called “apoptotic bodies”. These are taken up by nearby cells and phagocytes in order to prevent inflammation and tissue damage [54]. Apoptosis can be induced by two pathways: the intrinsic pathway, involving the mitochondria; and the extrinsic pathway, which relies on the activation of “death” receptors. Apoptosis is an ATP-dependent, enzyme-mediated, genetically programmed death of cells that are no longer necessary or pose a danger to the organism. This process also triggers the onset of many degenerative conditions associated with aging and oxidative stress [8,9,25,43,51,54,55,56,57,58,59,60,61,62].

One of the most relevant theories in this regard was raised in 1956 and addressed the impacts of free radicals in the aging process and oxidative damage associated with degenerative reactions. This hypothesis states that oxygen-derived free radicals are responsible for the age-associated impairment at the cellular and tissue layers [54,56]. Denham Harman was the first to suggest that mitochondria might be the biological clock in the aging process. Later, Miquel et al. proposed the mitochondrial theory of cell aging in 1980, arguing that cellular senescence signaling is induced by a by-product of oxyradicals attacking the mitochondrial genome of fixed postmitotic cells. Once mitochondria from postmitotic cells utilize a high percentage of O_2_, they release oxygen radicals, surpassing the capacity of the cellular antioxidant defense system [54,56]. The initial concepts proposed by Harman suggest that cellular senescent signaling is associated with OS. According to Helmut Sies, in senescent cells, there is an imbalance that favors pro-oxidants, which affects cell metabolism. This imbalance between those systems provokes the delivery of proapoptotic molecules and proteins by mitochondrial DNA (mtDNA) damage. Subsequently, the decompartmentalization process of these proapoptotic signaling compounds is initiated after mtDNA damage induced by the increase in ROS and RNS by-products [8,9,25,43,50,54,55,56,57,62].

All those mitochondrial electron transport chain complexes generate free radicals as a by-product [25,44,54,63]. These residues of mitochondrial activity have recently been shown to be involved in cellular signaling pathways, especially in the CNS. Oxidative Stress is characterized by the dysfunctional imbalance of the antioxidant defense mechanism and the exacerbated production of free radicals, including hydrogen peroxide, nitric oxide (NO), peroxynitrite, and a superoxide anion. A superoxide radical anion is an oxygen-free radical that interacts with hydrogen peroxide and NO to produce hydroxyl radicals [44,45,48,54,56,57,64]. Consequently, hydrogen peroxide can also interact with iron, through the Haber–Weiss reaction, generating more hydroxyl radicals. NO can also establish other reactions to form hydroxyl radicals [31,50,51,62,64].

DNA damage has been observed in a wide range of mammalian cell types when exposed to continuous or excessive oxidative stress by overloading free radical delivery. This oxidative damage to DNA can include single- and double-strand breaks, deletions, base changes, and even chromosomal aberrations [56,57]. The major molecular mechanisms involved in oxidative damage are the direct reactions of hydroxyl radicals and carbonyl compounds of DNA and nuclease activation. Hydroxyl radicals can potentially link and attack deoxyribose, purines, and pyrimidines, generating numerous byproducts intracellularly, like 8-hydroxydeoxyguanosin (8-oxodG), thymidine glycol, and 8-hydroxyadenosine. Mitochondrial DNA is the most susceptible to oxidative damage and mutations. Given the lack of protective histones in their composition, the free radicals produced by mitochondria increase the 8-oxodG formation in mtDNA via its degradation [12,34,42,44,45,50,54,64].

The apoptotic intrinsic pathway process begins after aggravated mtDNA damage, delivering not just apoptotic membrane signaling compounds but also cytochrome c [56,57]. When cytochrome c is delivered into the cytosol, it interacts with apoptotic protease-activating factor-1 (Apaf-1), ATP, and caspase9 forming an apoptosome. This apoptosome is aggregated by Apaf-1 being exposed to a caspase recruitment domain (CARD), recruiting pro-caspase9 to activate caspase9, and creating a proteolytic cleavage. In turn, caspase9 activates other caspases that result in cell death via proteolytic cleavage. The oxidative damage of mtDNA also releases apoptosis-inducing factor (AIF). Such inducing factor has the main function of translocation into the nucleus, causing cellular chromatin condensation and partial DNA fragmentation [54,55,56,57,64].

With regard to the neuronal microenvironment, which is rich in O_2_ due to the aerobic respiration demand of brain tissue, the cells and molecules of this specific system are rendered susceptible to oxidative stress. In fact, the aging process was associated with declining cognitive function and motor skills, such as the ones commonly seen in degenerative neurological conditions. These declines are also associated with oxidative protein damage within distinct areas of the brain [3,7,8,9,31,33,36,37,39,40,41,56,58,59,60,61,63,64,65].

### 4.5. Clinical Applications of Nebulized Glutathione in Neurodegenerative Conditions

The nebulization of glutathione offers a direct delivery method to the CNS, potentially enhancing its therapeutic efficacy in neurodegenerative conditions. Recent studies have explored its application in diseases such as PD and AD. Previous investigations [66,67,68] revealed that intranasal administration of glutathione can effectively elevate GSH levels in the brain without significant adverse effects. Elevated GSH levels in the CNS are responsible for neuroprotection and management of pathological progression.

Regarding the dose, in a randomized double-blind placebo-controlled study led by Mischley et al. [68], participants with PD were randomly assigned to receive either placebo (saline), 300 mg/day, or 600 mg/day of intranasal glutathione, divided into three daily doses. Follow-up assessments included screening for side effects related to PD symptoms and cognitive function, evaluating blood chemistry, monitoring for sinus irritation, and checking for reduced sense of smell. Tolerability was assessed based on the frequency and severity of reported adverse events, participant compliance, and any withdrawals from the study. Out of the 30 participants, 28 completed the study. One participant withdrew due to schedule conflicts and another withdrew due to adverse events attributed to the study medication, including a ringing sensation in the head and exacerbation of chronic pruritus. Compliance with the study medication met the criteria for tolerability in all cohorts. Commonly reported side effects included mild sinus symptoms, such as throat irritation and increased thirst, which were approximately equivalent across all of the study arms.

No significant differences in the frequency of adverse events were observed between the groups. Tolerability was confirmed by the frequency and severity of reported adverse events, compliance, and withdrawal rates. UPDRS (Unified Parkinson’s Disease Rating Scale) scores, included as a safety measure, showed improvement in both treatment arms compared to the placebo. Specifically, the 300 mg/day group showed a mean improvement of 5.3 points and the 600 mg/day group showed a mean improvement of 4.3 points. Other positive outcomes reported included an improved blink rate, reduced muscle pain, and improved arm swing. Laboratory tests, including complete blood count (CBC), alanine aminotransferase (ALT), aspartate aminotransferase (AST), creatinine, blood urea nitrogen (BUN), and urinalysis, showed no significant deviations from normal reference ranges across the groups. Specific improvements were noted in the UPDRS total scores and subscores, indicating a positive trend in symptom management with intranasal glutathione.

The study [68] supports the safety and tolerability of intranasal glutathione in individuals with PD. The findings suggest that intranasal glutathione may offer a non-dopaminergic therapeutic strategy to improve PD symptoms, warranting further research through larger more comprehensive clinical trials. The Unified Parkinson’s Disease Rating Scale is a comprehensive tool used to assess the severity and progression of PD [69,70]. It consists of four parts: Part I evaluates mentation, behavior, and mood; Part II assesses activities of daily living (ADL); Part III focuses on motor examination; and Part IV addresses complications [69,70]. Each part includes various items rated on a scale from 0 to 4, with higher scores indicating greater severity of symptoms. The total score can range from 0 to 199. In the context of the study led by Mischley et al. [68], the UPDRS was used to monitor changes in PD symptoms to evaluate the efficacy of intranasal glutathione. Improvements in UPDRS scores suggest a reduction in the severity of PD symptoms. Specifically, the 300 mg/day group showed a mean improvement of 5.3 points, while the 600 mg/day group showed a mean improvement of 4.3 points compared to the placebo group [68]. These improvements indicate a notable reduction in symptom severity, suggesting that intranasal glutathione may effectively alleviate PD symptoms, thereby enhancing patients’ quality of life and functional abilities. The UPDRS is crucial in PD clinical trials as it provides a standardized measure to compare changes in disease severity and patient outcomes.

These results highlight the potential benefits of intranasal glutathione as a safe and effective treatment option for managing symptoms of Parkinson’s Disease, with a favorable safety profile and notable improvements in clinical outcomes. Additional pharmacokinetic and dose-finding studies are still recommended to further optimize and support its therapeutic use in regenerative medicine.

While there is currently a lack of RCTs specifically evaluating the effects of nebulized glutathione in AD, it is important to recognize the similarities in the pathological progression and underlying mechanisms of both AD and PD. Both neurodegenerative disorders are characterized by significant oxidative stress, mitochondrial dysfunction, and neuroinflammation, contributing to neuronal degeneration and disease progression. The successful application of intranasal glutathione in PD, evidenced by improvements in oxidative stress markers and symptom management, provides a strong rationale for exploring its potential benefits in AD. Given the shared pathways of neurodegeneration, including the role of ROS and the impairment of the NRF2 pathway, it is plausible that the therapeutic effects observed in PD could extend not only to AD but also to other brain disorders.

Moreover, ongoing research is evaluating the nasal route as an alternative pathway that allows drugs to be directly delivered to the brain via the nasal cavity [71]. For instance, Cunha et al. [71] explored the nose-to-brain delivery of therapeutic agents for AD using advanced formulations such as nanoemulsions, nanostructured lipid carriers, and in situ hydrogels, demonstrating the potential of these methods to enhance drug efficacy in AD. This study emphasizes that nasal delivery can bypass the BBB, providing a non-invasive route to deliver therapeutic agents directly to the brain. Such innovative approaches reinforce the potential of nebulized glutathione as a promising intervention for AD, leveraging the insights gained from PD studies and the benefits of advanced nasal delivery systems to inform and design future clinical trials aimed at addressing oxidative stress and neuroinflammation inherent in both conditions.

### 4.6. Author’s Note

The increasing prevalence of neurodegenerative disorders such as Parkinson’s Disease and Alzheimer’s Disease, coupled with the significant role oxidative stress plays in their pathogenesis, underscores the urgent need for innovative therapeutic approaches.

We propose that nebulized glutathione offers a novel and promising method for delivering this potent antioxidant directly to the respiratory system and subsequently the CNS, potentially enhancing its bioavailability and therapeutic efficacy. By bypassing the gastrointestinal tract, nebulization may provide a more direct and efficient means of increasing GSH levels in tissues, especially the brain, which is highly vulnerable to oxidative damage.

The literature reveals a growing body of evidence supporting the role of glutathione in mitigating oxidative stress, preserving mitochondrial function, and protecting neuronal health. The potential benefits of nebulized GSH include enhanced NRF2 pathway signaling, improved mitochondrial function, and overall neuroprotection, which is critical in combating the progressive neurodegeneration seen in these disorders.

While the preliminary findings are promising, further clinical research is warranted to fully elucidate the therapeutic potential of nebulized glutathione. Rigorous clinical trials and studies are necessary to determine the optimal dosing, safety, and long-term efficacy of this intervention. We advocate for continued research and collaboration in this field to explore the full spectrum of benefits that nebulized glutathione may offer to patients suffering from neurodegenerative conditions.

## 5. Conclusions

Glutathione plays an indispensable role in the endogenous antioxidant system and in maintaining redox balance during mitochondrial ATP synthesis. Neurodegenerative conditions feature defects in both NRF2-mediated antioxidant defenses and mitochondrial electron transport chain activity, which may be ameliorated through glutathione repletion.

Research indicates that glutathione-centered therapies, including nebulization, hold promise for mitigating neuronal oxidative damage and bioenergetic failure underlying neurodegeneration. More specifically, glutathione is vital for cellular health, especially in mitochondria. It neutralizes harmful reactive oxygen species produced during electron transport, protecting essential proteins and coenzyme Q. Glutathione also maintains vitamin E in its active form, guarding mitochondrial membranes from oxidative damage. It supports detoxification enzymes, aiding in the metabolism of toxins. Glutathione may also help preserve mitochondrial DNA integrity.

Overall, it plays a crucial role in maintaining mitochondrial health and efficient electron transport in cells. Further clinical evaluation of glutathione-boosting protocols is warranted.

## Figures and Tables

**Table 1 nutrients-16-02476-t001:** Detailed biochemical properties and functions of glutathione.

Biochemical Property	Description	Significance
Composition	Tripeptide consisting of cysteine, glycine, and glutamate	Sulfhydryl group (-SH) from cysteine contributes to its antioxidant properties
Synthesis	Occurs in two ATP-dependent steps involving glutamate-cysteine ligase (GCL) and GSH synthase (GS)	Rate-limited by the availability of cysteine; occurs primarily in the cytosol and distributed to various organelles
GSH-GSSG Cycle	Involves the reduction of GSSG to GSH by glutathione reductase (GR) and oxidation of GSH to GSSG by glutathione peroxidase (GPx)	Essential for detoxification of hydrogen peroxide (H_2_O_2_) and lipid peroxides, protecting cells from oxidative damage
Cellular Distribution	Found in the cytosol, mitochondria, endoplasmic reticulum, and nucleus	Indicates its universal role in cellular defense mechanisms and in regulating cellular events such as apoptosis, autophagy, and mitochondrial function
Role in Neuroprotection	Especially crucial in the brain due to its vulnerability to oxidative stress	Protects neurons by maintaining mitochondrial function, regulating neurotransmitter release, and modulating the inflammatory response

**Table 2 nutrients-16-02476-t002:** Glutathione’s role in NRF2 pathway activation and antioxidant response.

Component	Function in NRF2 Pathway	Impact on Antioxidant Defense
NRF2	Transcription factor that activates the antioxidant response	Increases expression of antioxidant enzymes including GPx and GST
Keap1	Sensor protein that binds NRF2 in the cytoplasm, targeting it for degradation	Oxidative stress modifies Keap1, releasing NRF2 to enter the nucleus
ARE (Antioxidant Response Element)	DNA sequence in the promoter region of antioxidant genes	Binding site for NRF2, initiating transcription of antioxidant genes
Glutathione S-transferases (GSTs)	Enzymes that catalyze the conjugation of GSH to electrophilic compounds	Protect cells from oxidative damage and support detoxification processes
Glutaredoxins (Grx)	Small redox proteins involved in reversible protein glutathionylation	Contribute to redox regulation and signal transduction under oxidative stress

**Table 3 nutrients-16-02476-t003:** The role of glutathione in the pathological features of PD and AD.

Pathological Feature	Parkinson’s Disease (PD)	Alzheimer’s Disease (AD)	Role of Glutathione (GSH)
Oxidative Stress	High levels of reactive oxygen species (ROS)	Elevated oxidative stress markers	GSH neutralizes ROS, reducing oxidative damage
Mitochondrial Dysfunction	Impaired mitochondrial function, decreased ATP production	Mitochondrial abnormalities, disrupted energy metabolism	GSH supports mitochondrial function and energy production
Neuroinflammation	Chronic inflammation in the substantia nigra	Neuroinflammatory processes in the cortex and hippocampus	GSH reduces inflammation by modulating the NRF2 pathway
Protein Aggregation	Accumulation of alpha-synuclein	Accumulation of amyloid-beta plaques	GSH aids in the degradation of misfolded proteins and aggregates
Neuronal Loss	Loss of dopaminergic neurons in the substantia nigra	Loss of cholinergic neurons in the cortex and hippocampus	GSH protects neurons by mitigating apoptosis and autophagy
Impaired Neurotransmission	Dopamine deficiency leading to motor symptoms	Acetylcholine deficiency leading to cognitive decline	GSH helps maintain neurotransmitter balance by protecting neuronal health

**Table 4 nutrients-16-02476-t004:** Detailed overview of glutathione in mitochondrial function.

Mitochondrial Aspect	Function of Glutathione	Importance
Electron Transport Chain (ETC)	GSH protects components of the ETC, particularly complex I and II	Ensures efficient ATP production by minimizing oxidative damage to ETC proteins
ROS Neutralization	Directly neutralizes ROS, reducing oxidative stress	Prevents damage to mitochondrial DNA, lipids, and proteins
GSH Cycle in Mitochondria	Facilitates the reduction of GSSG back to GSH within mitochondria	Maintains a high GSH:GSSG ratio, critical for redox balance and cellular health
Detoxification	GSH acts as a cofactor for detoxification enzymes in mitochondria	Supports the metabolism and elimination of toxins and xenobiotics
Protection of mtDNA	GSH may help preserve the integrity and function of mitochondrial DNA	Essential for preventing mutations and ensuring the continuity of mitochondrial function

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
