# Peer review of "Nebulized Glutathione as a Key Antioxidant for the Treatment of Oxidative Stress in Neurodegenerative Conditions"

_nutrients, 2024, doi:10.3390/nu16152476_

Round 1

Reviewer 1 Report

Comments and Suggestions for Authors

The present submission is a well-organized literature survey of the biosynthesis and the biochemical roles of glutathion. Contrary to the title, however, no details (even a reference) are mentioned relating to the nebulized glutathione, its clinical application, its possible side effects, etc.

Therefore, I wish to ask you to ask the authors to complete the literature survey, reduce the „standard” biochemistry parts of the present form, and concentrate on the application of nebulized glutathione in the particular diseases mentioned in the manuscript.

Reviewer 2 Report

Comments and Suggestions for Authors

1. The authors' paper is a review on the antioxidant effects of nebulized glutathione. I am curious if there are any preliminary research results or references related to nebulized glutathione. Additionally, I am interested in the authors' thoughts on how they came up with the concept of nebulized glutathione.

2, It seems that there are too few references related to glutathione in Parkinson's disease and Alzheimer's disease. This may indicate a lack of evidence for the therapeutic effect of nebulized glutathione aspiration. What are the authors' thoughts on this?

3. As a review paper providing information on new nebulized therapy, it would be easier to understand if the results and interpretations from various references about the antioxidant effects and anti-apoptotic effects of nebulized therapy or glutathione in neurological disorders were described using diagrams and tables. What are the authors' thoughts on this?

Reviewer 3 Report

Comments and Suggestions for Authors

Lana et al. wrote an article about glutathione and its role in neurodegenerative conditions. The article is comprehensive, and well-written, the topic is particularly relevant today. The article provides sufficient and satisfactory information on the subject.

However, some questions came to my mind after reading:

1. Since the article's title suggests the role of glutathione in neurodegenerative disorders, these conditions should be more detailed. For example, the article should contain subsections about each condition e.g. Alzehimer, Parkinson, Dementia, ALS, SM.

2. Creating figures would further increase the value of the article. Subsections e.g. glutathione biochemistry would be better understood with figures.

3. Minor editing of English language required.

Comments on the Quality of English Language

3. Minor editing of English language required.

Round 2

Reviewer 1 Report

Comments and Suggestions for Authors

The authors modified the submission according to the criticism in the Review Report.